# Introduction of 24 h Resident Consultant Cover in a Tertiary Neonatal Unit—Impact on Mortality and Clinical Outcomes

**DOI:** 10.3390/children8100865

**Published:** 2021-09-28

**Authors:** Carolina Zorro, Theodore Dassios, Ann Hickey, Anusha Arasu, Ravindra Bhat, Anne Greenough

**Affiliations:** 1Neonatal Intensive Care Centre, King’s College Hospital NHS Foundation Trust, London SE5 9RS, UK; carolina.zorro@nhs.net (C.Z.); annm_hickey@hotmail.com (A.H.); anusha.arasu@nhs.net (A.A.); ravindra.bhat@nhs.net (R.B.); 2Department of Women and Children’s Health, School of Life Course Sciences, Faculty of Life Sciences and Medicine, King’s College London, London SE5 9RS, UK; anne.greenough@kcl.ac.uk; 3Asthma UK Centre for Allergic Mechanisms in Asthma, King’s College London, London SE5 9RS, UK; 4NIHR Biomedical Research Centre, Guy’s and St Thomas’ NHS Foundation Trust and King’s College London, London SE1 9RT, UK

**Keywords:** resident consultants, neonatal outcomes, mortality

## Abstract

Background: We aimed to determine whether the introduction of 24 h cover by resident consultants in a tertiary neonatal unit affected mortality and other clinical outcomes. Methods: Retrospective cohort study in a tertiary medical and surgical neonatal unit between 2010–2020 of all liveborn infants admitted to the neonatal unit. Out of hours cover was rearranged in 2014 to ensure 24 h presence of a senior trained neonatologist (resident consultant). Results: In the study period, 4778 infants were included: 2613 in the pre-resident period and 2165 in the resident period. The median (IQR) time to first consultation by a senior member of staff was significantly longer in the pre-resident period [1.5 (0.6–4.3) h] compared to the resident period [0.5 (0.3–1.5) h, *p* < 0.001]. Overall, mortality was similar in the pre-resident and the resident periods (3.2% versus 2.3%, *p* = 0.077), but the mortality of infants born at night was significantly higher in the pre-resident (4.5%) compared to the resident period (2.5%, *p* = 0.016). The resident period was independently associated with an increased survival to discharge (adjusted *p* < 0.001, odds ratio: 2.0) after adjusting for gestational age, admission temperature and duration of ventilation. Conclusions: Following introduction of a resident consultant model the mortality and time to consultation after admission decreased.

## 1. Introduction

Adverse neonatal outcomes are associated with increased mortality and long term disability [1]. Admission to a neonatal unit, however, is a common event with approximately one in ten infants being admitted in a high-income setting. The rate of admissions to neonatal care is increasing in the current era and is only expected to rise further in the near future [2].

The traditional model of neonatal care in the United Kingdom consisted of resident consultant clinical cover during standard working hours (8 am to 5 pm) with the on-call consultant covering the service from home during late evening and at night. In a 24 h consultant-led service though, the consultant is clinically responsible for patient care and will provide either hands-on care or closely supervise in the clinical setting all aspects of care. This model of care requires a consultant to be present in the hospital outside normal working hours and hence the term ‘resident consultant’ refers to physicians working within this model most typically on shift patterns [3].

The British Association of Perinatal Medicine has recommended 24 h availability of a consultant neonatologist for tertiary neonatal units [4]. Similar recommendations have been made to improve the quality of care given to infants in neonatal units by the Royal College of Paediatrics and Child Health [3] and the Department of Health [5]. Although, these recommendations have been in place for a number of years and in the recent years some units in the United Kingdom have introduced resident consultant cover, the effect of this intervention on neonatal outcomes has not been evaluated in the UK [6]. Studies from Australia [7] and Canada [8] have reported no significant differences in neonatal or neurodevelopment outcomes at three years of age, but the studies were relatively small and their results might not be directly applicable to a different health system.

Our aim was to compare mortality and other outcomes before and after the introduction of a resident consultant model in a UK tertiary medical and surgical neonatal intensive care unit.

## 2. Materials and Methods

### 2.1. Subjects and Study Design

A retrospective cohort study of all admissions to the Neonatal Intensive Care Centre at King’s College Hospital (KCH) between 1 January 2010 and 1 January 2020 was undertaken. KCH has a tertiary neonatal unit with approximately 6000 deliveries per year and serves a diverse community of over 1,000,000 in southeast London. Data were extracted from the BadgerNet Neonatal Electronic Patient Database (Clevermed, Edinburgh, UK). If an infant had multiple admissions to the unit following referral from a local hospital or repatriation following specialist care, only the first episode was included in the analysis. Mortality was defined as death before discharge from neonatal care [9]. The following data were collected: maternal age (years), any antenatal steroids (yes/no), cord arterial pH, inborn at KCH (yes/no), gestational age (weeks), birth weight (kg), sex (male/female), Apgar score at 10 min, time of admission, admission temperature (°C), admission blood glucose (mmol/L), time to first consultation by a senior member of staff (in hours), mechanical ventilation (yes/no), duration of invasive ventilation (days), duration of supplementary oxygen (days), duration of parenteral nutrition (PN) (days), duration of intensive care (days [10], discharge home on supplemental oxygen (yes/no), intraventricular haemorrhage (IVH) grade III-IV or periventricular leucomalacia (PVL) (yes/no), postmenstrual age at discharge (weeks), weight at discharge (kg).

### 2.2. Derived Outcomes and Composite Parameters

Night mortality was defined as any death before discharge from neonatal care of infants that were admitted during the night (8:00 pm to 8:00 am) and day mortality was defined as any death before discharge from neonatal care of infants that were admitted during the day (8:00 am to 8:00 pm). The same time frames were used to classify admission hypothermia into day or night hypothermia. The birthweight z-score was calculated using the UK-WHO preterm reference chart [11] and the Microsoft Excel add-in LMSgrowth (V.2.77; www.healthforchildren.co.uk, accessed on 7 June 2021). The time of admission was used to group admissions to day (any admission between 8:00 am and 8:00 pm) or night (any admission from 8:00 pm to 8:00 am). Hypothermia was defined as an admission temperature of <36.5 °C [12]. Hypoglycaemia was defined as a blood glucose concentration of <2.6 mmol/L [13].

### 2.3. Resident Consultants

The clinical cover at night before the introduction of the resident consultants consisted of a non-resident consultant, a senior trainee usually at the last three years of neonatal training and two junior trainees. Out of hours cover was rearranged to ensure 24 h presence of a senior trained neonatologist (resident consultant). The model was partly introduced in March 2014 with complete implementation by September 2015. By September 2015 a total of six resident consultants were appointed to form a separate tier of cover at night which was additional to the medical staff of the previous model. The period of implementation and a washout period of the following full chronological year (2016) were excluded from the analysis. The washout period was deemed essential as in 2016 not all posts were consistently filled. The pre-resident period thus corresponded to the complete years 2010–2013 (4 years) and the resident period to 2017–2019 (3 years). The year 2020 was not included as not all infants were discharged from neonatal care when this manuscript was written and because of the impact of the COVID-19 pandemic on the medical workforce resulting in redeployment to other clinical areas. The total number of beds did not change during the study period.

### 2.4. Statistical Analysis

The data were tested for normality with the Kolmogorov Smirnoff tests and were found to be not normally distributed and were thus presented as median (interquartile range). The Mann–Whitney U test was used for comparisons of variables before and after the implementation of the resident consultant model. Binary variables were compared between the two periods with the x^2^ test. The incidence of male sex, gestational age, birth weight z-score, admission hypothermia and duration of ventilation were compared between the infants that survived to discharge versus the infants that did not. A multivariable binary logistic regression model with survival to discharge from neonatal care was created to examine the independent association of the resident period on survival to discharge after correcting for confounding parameters that were different (*p* < 0.1) between the infants who survived to discharge versus the ones that did not survive. Multi-collinearity among the independent variables in the regression analysis was assessed by examining a correlation matrix for the independent variables. Statistical analysis was performed using SPSS V.26.0 software.

## 3. Results

A total of 6867 infants were admitted to the Neonatal Intensive Care Centre at KCH in the study period. During the pre-resident period 2613 infants were admitted, and in the resident period 2165 infants were admitted. In the implementation and washout period a total of 2089 infants were admitted and excluded from the analysis.

The incidence of admission hypothermia was significantly higher in the pre-resident period (31%) compared to the resident period (15%, *p* < 0.001).

The median (IQR) time to first consultation was significantly longer in the pre-resident period [1.5 (0.6–4.3) hours] compared to the resident period [0.5 (0.3–1.5) h, *p* < 0.001) (Table 1).

In the pre-resident period 6.6% of admissions were born at less than 28 weeks of gestation compared to 8.3% in the resident period (*p* = 0.023) (Table 1). The admitted infants did not differ significantly in the two periods in birth weight, birth weight z score, sex or Apgar score at 10 min. Fewer infants in the pre-resident period (26.4%) were ventilated compared to the resident period (38.9%, *p* < 0.001) and fewer infants were discharged on home oxygen in the pre-resident period (7.0%) compared to the resident period (11.6%, *p* < 0.001). The incidence of IVH grade III-IV or PVL was not significantly different in the pre-resident period (25.6%) compared to the resident period (33.3%). The overall mortality was 3.2% in the pre-resident period and 2.3% in the resident period (*p* = 0.077) (Table 1, Figure 1). The night mortality was significantly higher in the pre-resident period (4.5%) compared to the resident period (2.5%, *p* = 0.016) (Table 1, Figure 1).

The infants who died before discharge from neonatal care did not differ in the incidence of male sex (79 of 133, 59.4%) compared to the incidence of male sex in the infants that survived (2616 of 4645, 56.3%, *p* = 0.723) (Table 2).

The infants who died before discharge from neonatal care were of a smaller gestational age [32.9 (25.1–38.3) weeks] and smaller birth weight z-score [−0.53 (−1.03–0.14)] compared to the ones that survived [37.4 (33.7–39.9) weeks, *p* < 0.001 and −0.35 (−1.06–0.37), *p* = 0.064, respectively]. The infants that died before discharge from neonatal care had a longer duration of invasive ventilation [4 (2–14) days] and lower admission temperature [36.5 (36.1–36.9) °C] compared to the infants that survived [2 (1–6) days, *p* < 0.001 and 36.8 (36.5–37.0) °C, *p* < 0001, respectively]. Following multivariable binary regression analysis, the resident period was independently associated with increased survival to discharge from neonatal care (adjusted *p* < 0.001, odds ratio: 2.0, 95% CI:1.4–3.1) after adjusting for differences in gestational age, birth weight z score, admission temperature, days of ventilation and admission during the night hours (Table 3).

## 4. Discussion

We have demonstrated that mortality, hypothermia and time to consultation by a senior staff member decreased during a period of resident consultants. The time period with resident consultants was associated with increased survival to discharge from neonatal care.

Over a ten-year period, a number of changes have been introduced to neonatal care that might partially explain the improved outcomes in our study. Such major interventions include volume targeted ventilation [14], the introduction of high flow nasal cannula therapy [15], the application of the current transcutaneous saturation targets [16] and the universal use of whole body hypothermia for hypoxic ischemic encephalopathy [17]. There is a world-wide tendency for improved survival of extremely preterm infants and a corresponding increase in the rate of prematurity-related complications such as chronic lung disease [18]. In our study, though, we report improved night mortality compared to relatively unchanged day mortality which is likely to reflect improved out of hours clinical care following senior medical presence, as the day level of cover was largely unchanged for the study period. It is also important to note, that by “night mortality” we do not refer to infants who died at night, but infants that were initially resuscitated and stabilised at night and died later during their stay in the unit. This finding is in agreement with the previously described observation that events that happen during or shortly after birth have a significant and long-lasting effect on later outcomes and underline the importance of the golden hour of care in neonatal services [19]. We do appreciate, however, the limitations of splitting mortality in day and night as even for a baby that is born during the day time, the care would still be provided continuously over both day and night.

Resnick and colleagues retrospectively studied 480 infants born at less than 32 weeks of gestational age before and after adopting an after-hours in-house senior physician cover and reported no major differences in neonatal outcomes [7]. Lodha and colleagues retrospectively studied 387 preterm infants before and after the introduction of a 24 h in-house staff neonatologist cover and reported no significant difference in neurodevelopmental outcomes at three years corrected age [8]. Our differing results of better outcomes following the introduction of resident consultants might be explained by population differences as we conducted a larger study and included all infants, rather than only those born prematurely. The different time periods might also explain the differences in our results as Lodha et al. examined infants born between 1998 and 2004. In the study of Resnick et al., consultants attended high risk deliveries even before the introduction of the resident model, while the senior resident neonatologists were not only consultants but “consultant or neonatal fellow”.

It is interesting to note that the rate of admissions at night did not decrease after the introduction of the resident model. This is in contrast to a study in paediatric emergency medicine where the presence of resident consultants was associated with reduced night-time admissions [20]. This might be explained by the relatively better-defined criteria for admission to neonatal care and that newborn infants already constitute a high risk population, that is already in hospital (labour or postnatal ward) before admission to a neonatal unit. The activity of our unit increased during the study period with an increasing proportion of extremely preterm infants and longer durations of invasive ventilation. It follows thus that the expectation would be for mortality to increase rather than decrease, an observation which strengthens the conclusions of our study.

It is important to note that a number of our included outcomes such as early first communication from a senior team member and the improved mortality and admission hypothermia are also recognised neonatal service quality indicators and their improvement could be seen as a reflection of continuous quality improvement in the healthcare of the newborn [19]. The improvement in admission hypothermia, other than the resident model, might also be attributed to increased awareness of the complications associated with hypothermia in newborn infants. The International Liaison Committee on Resuscitation concluded in 2015 that admission temperature of newborn infants was a strong predictor of mortality and morbidity at all gestations and published a consensus statement that temperature should be maintained between 36.5 °C and 37.5 °C after birth through stabilisation and admission [21]. Following this, and since 2015, the British national neonatal audit programme has specifically asked the question if an admitted infant had a first measured temperature within one hour of birth.

Our study has strengths and some limitations. This is, to our knowledge, the first neonatal study from the UK conducted in a tertiary medical and surgical neonatal unit that quantified the effect of resident consultant cover on neonatal outcomes. We used a large population of infants that were cared for at the same unit minimising potential differences that relate to different clinical practices or standards of care between units. The applicability of our study lies in that the improved neonatal outcomes following the introduction of resident consultants, if confirmed by larger and more diverse-population studies, could strengthen the argument of implementing a resident out-of-hours model of care. Since the electronic patient records software was introduced only shortly before the beginning of our study period in 2010, some less-straightforward diagnoses were inconsistently entered in the early pre-resident period. For example, diseases such as bronchopulmonary dysplasia that require an assessment at 28 days or 36 weeks postmenstrual age, or culture-negative sepsis and suspected necrotising enterocolitis were not always recorded in the early pre-resident period of the study, but were consistently entered at the later-resident period. Had we included these items, we would have presented an artificially higher incidence in the later period which is mostly due to better data entry rather than a truly higher incidence of these conditions. We chose, thus, not to present these items in our manuscript. We should acknowledge as a limitation that it is impossible to completely separate the contribution of the resident consultants to improved outcomes from the overall tendency for better outcomes in neonatal care. Our improved “night outcomes” though point towards improved out of hours care by the resident consultants. Our study, however, clearly cannot establish a clear causal link between the introduction of residents and improved neonatal outcomes as it is retrospective and spans over a decade. Nevertheless, it is important to share with the neonatal community our experience of this costly intervention which has significant policy implications.

## 5. Conclusions

In conclusion, introduction of resident consultants was associated with a positive impact on reducing neonatal mortality, the incidence of hypothermia and time to consultation after admission. We suggest that those planning neonatal services could consider adopting a similar model.

## Figures and Tables

**Figure 1 children-08-00865-f001:**
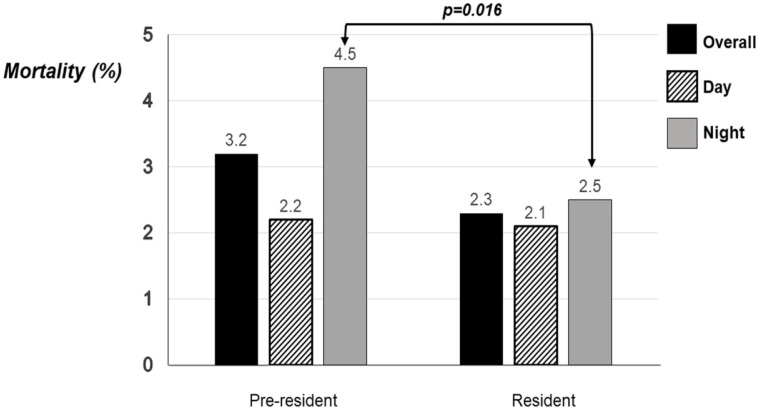
Total, day and night mortality before discharge from neonatal care during the pre-resident and resident periods.

**Table 1 children-08-00865-t001:** Characteristics and outcomes of the admitted infants in the two study periods. Median (IQR) or *N* (%). Comparisons by Mann–Whitney U or x^2^ test as appropriate.

	Pre-Resident *N* = 2613	Resident *N* = 2165	*p* Value
Maternal age (years)	31 (27–35)	32 (28–36)	<0.001
Antenatal steroids (in <34 weeks)	597 of 679 (88)	551 of 594 (93)	0.015
Cord arterial pH	7.24 (7.16–7.30)	7.22 (7.13–7.29)	0.001
Inborn at KCH	2306 (88)	1740 (80)	<0.001
Gestational age (weeks)	37.3 (33.7–39.7)	37.6 (33.3–40.0)	0.304
Gestation < 28 weeks	172 (6.6)	180 (8.3)	0.023
Birth weight (kg)	2.66 (1.86–3.37)	2.76 (1.77–3.41)	0.518
Birth weight z score	−0.37 (−1.09–0.37)	−0.34 (−1.02–0.35)	0.705
Male sex	1452 (56)	1243 (57)	0.069
Apgar at 10 min	10 (9–10)	10 (9–10)	0.646
Admissions at night	1102 (42.2)	945 (43.6)	0.305
Admission hypothermia	816 (31)	324 (15)	<0.001
Day admission hypothermia	489 of 1505 (32.5)	187 of 1220 (15.3)	<0.001
Night admission hypothermia	327 of 1091 (30.0)	137 of 944 (14.5)	<0.001
Admission hypoglycaemia	842 (32.2)	531 (24.5)	<0.001
Time to first consult (h)	1.5 (0.6–4.3)	0.5 (0.3–1.5)	<0.001
First consultation > 6 h	421 (16.1)	151(7.0)	<0.001
Mechanical ventilation	691 (26.4)	843 (38.9)	<0.001
Days of ventilation (ventilated only)	3 (1–6)	3 (1–7)	0.496
Days of oxygen (in the ones that had oxygen, *N* = 2179)	2 (1–4)	3 (1–9)	<0.001
Days of PN (in the ones that had PN, *N* = 1332)	10 (6–18)	12 (7–22)	<0.001
Days of intensive care (per year)	1256	3940	N/A
Home oxygen	184 (7.0)	252 (11.6)	<0.001
IVH grade III-IV or PVL (% of <28 weeks)	44 (25.6)	60 (33.3)	0.239
Overall mortality	83 (3.2)	50 (2.3)	0.077
Day admissions mortality	33 of 1511 (2.2)	26 of 1220 (2.1)	0.925
Night admissions mortality	50 of 1102 (4.5)	24 of 945 (2.5)	0.016
Postmenstrual age at discharge (weeks)	38.6 (36.4–41.0)	39.2 (36.3–41.1)	0.568
Weight at discharge	2.80 (2.10–3.43)	2.93 (2.12–3.50)	0.050

**Table 2 children-08-00865-t002:** Comparison of infants that survived versus infants that died before discharge from neonatal care. Median (IQR) or *N* (%). Comparisons by Mann–Whitney U or x^2^ test as appropriate.

	Died *N* = 133	Survived *N* = 4645	*p* Value
Male sex	79 (59.4)	2616 (56.3)	*p* = 0.723
Gestational age (weeks)	32.9 (25.1–38.3)	37.4 (33.7–39.9)	*p* < 0.001
Birth weight z-score	−0.53 (−1.03–0.14)	−0.35 (−1.06–0.37)	*p* = 0.064
Duration of ventilation (days)	4 (2–14)	2 (1–6)	*p* < 0.001
Admission temperature (°C)	36.5 (36.1–36.9)	36.8 (36.5–37.0)	*p* < 0001

**Table 3 children-08-00865-t003:** Binary logistic regression analysis with survival to discharge from neonatal care as the outcome variable.

	Adjusted *p*	Odds Ratio	95% Confidence Intervals
Resident period	<0.001	2.02	1.40–3.11
Gestational age	<0.001	1.14	1.10–1.18
Birth weight z-score	0.288	1.10	0.92–1.31
Admission hypothermia	<0.001	1.72	1.39–2.12
Duration of ventilation	<0.001	0.98	0.97–0.99
Admission during the night	0.047	0.69	0.48–0.88

## Data Availability

Data can be made available upon request.

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
