# Peer review of "Introduction of 24 h Resident Consultant Cover in a Tertiary Neonatal Unit—Impact on Mortality and Clinical Outcomes"

_children, 2021, doi:10.3390/children8100865_

Round 1

Reviewer 1 Report

This is a retrospective cohort study in a tertiary medical and surgical neonatal unit between 2010-2020 consisting of 4,778 infants and is designed to determine whether the introduction of 24-hour cover by resident consultants in a tertiary neonatal unit affected mortality and other clinical outcomes. After analysis, the authors concluded that following introduction of a resident consultant model the mortality and time to consultation after admission decreased.

Comments:

Regarding this issue, several situations should be discussed.

  1. Was there any change in terms of guidelines or protocols before and during resident consultation? Particularly, was there any change of management to prevent admission hypothermia?
  2. Was there any change in terms of manpower between two different periods?
  3. Was there any change of beds during these two different periods? It is important to describe the volume or capacity of patient care between these two periods.

Night or day mortality is not practical unless major events such as severe IVH, pneumothorax for preterm infants occur immediately during the admitted night. Even a newborn infant was admitted in the daytime, they received continuous treatment day and night.

It is also important to know the whole picture regarding quality of care, such as necrotizing enterocolitis, late-onset sepsis, bronchopulmonary dysplasia, or retinopathy of prematurity during these two periods.

Besides, it is good to know numbers of inborn or outborn patients during these two periods.

Author Response

Please find the responses attached as a word document.

Reviewer 2 Report

The question asked is interesting and the data at hand seems to be able to shed light on it. The statistical analysis and especially the presentation of the results can however be improved.

  1. In the section "statistical analysis" it would be nice to see a explanation of the function of the methods presented to answer the question in the paper. It seems to me that the centre piece in the study is the binary regression. (Here you could be more specific e.g. logistic regression.) The other tests are more for the descriptive tables. 
  2. The main result stemming from the regression is only mentioned in one sentence in the results. 168-172. It really deserves a full presentation in the paper with tables and so on.  
  3. How is the day/night difference handled in the regression?
  4. In the descriptive tables be more specific from what test the p values come from.
  5. Figure 1 is not correctly referred to in the text. (row 128-129)
  6. The sections are not correctly numbered. (row 59, 78, 91,107)

Author Response

(The authors gave the same response as above.)

Round 2

Reviewer 2 Report

I don't full understand the answer to comment 10

Comment 10: How is the day/night difference handled in the regression?

Response: The day/night difference was not inserted as a separate variable in the regression model as it is part of the “resident/non-resident” variable which is already part of the model. In order to account for day/night we would have needed another two separate regression analyses further to the one we have already performed, which might have rendered the results difficult to follow for the general readership.

I had interpret the resident non-resident variable as a period variable dividing it by time in pre/post period. But it is not? How can the night variable be included in the resident variable? What the variables are should be clarified. If it is a period variable the night admission variable could easily be included as a cross term. You don't need more than one regression. I think this should make the results more accessible and convincing for the reader. The main conclusion is that the resident period is beneficial for survival due to increased survival for night admissions. It would be really beneficial for the paper to include a cross term in the logistic regression to see how it holds controlled for the other variables. Now the night admission result is found by searching in Table 1. If results are presented good I think the general readership will understand. 

The conclusion: "The time period with resident consultants was independently associated with a twofold increase in the
chances of survival to discharge from neonatal care." is strange. What does this mean? If the chance of survival is 97% what is a twofold increase of the chances of survival? Take care to state the main conclusion correct.

Author Response

Comment 1: I don't fully understand the answer to comment 10

Comment 10: How is the day/night difference handled in the regression? Response: The day/night difference was not inserted as a separate variable in the regression model as it is part of the “resident/non-resident” variable which is already part of the model. In order to account for day/night we would have needed another two separate regression analyses further to the one we have already performed, which might have rendered the results difficult to follow for the general readership.

I had interpret the resident non-resident variable as a period variable dividing it by time in pre/post period. But it is not? How can the night variable be included in the resident variable? What the variables are should be clarified. If it is a period variable the night admission variable could easily be included as a cross term. You don't need more than one regression. I think this should make the results more accessible and convincing for the reader. The main conclusion is that the resident period is beneficial for survival due to increased survival for night admissions. It would be really beneficial for the paper to include a cross term in the logistic regression to see how it holds controlled for the other variables. Now the night admission result is found by searching in Table 1. If results are presented good I think the general readership will understand. 

Response: Thank you for further evaluating our work. We have now included “admission during the night” as a new item in the binary logistic regression and adjusted the values of the odds ratios and corresponding confidence intervals in the table for all the variables.

Comment 2: The conclusion: "The time period with resident consultants was independently associated with a twofold increase in the chances of survival to discharge from neonatal care." is strange. What does this mean? If the chance of survival is 97% what is a twofold increase of the chances of survival? Take care to state the main conclusion correct.

Response: We have now altered to “The time period with resident consultants was associated with increased survival to discharge from neonatal care.”
